# Screening of Tumor Antigens and Construction of Immune Subtypes for mRNA Vaccine Development in Head and Neck Squamous Cell Carcinoma

**DOI:** 10.3390/biom13010090

**Published:** 2022-12-31

**Authors:** Hong-Xia Li, Tian-Run Liu, Zhao-Xu Tu, Chu-Bo Xie, Wei-Ping Wen, Wei Sun

**Affiliations:** 1Department of Otorhinolaryngology Head and Neck Surgery, The Sixth Affiliated Hospital, Sun Yat-sen University, Guangzhou 510655, China; 2Guangdong Institute of Gastroenterology, The Sixth Affiliated Hospital, Sun Yat-sen University, Guangzhou 510655, China; 3Department of Otolaryngology, The First Affiliated Hospital, Sun Yat-sen University, Guangzhou 510655, China; 4Otorhinolaryngology Institute, Sun Yat-sen University, Guangzhou 510655, China

**Keywords:** head and neck squamous cell carcinoma, mRNA vaccine, tumor antigen, immune subtypes, immune landscape

## Abstract

Background: A growing number of clinical studies have confirmed that mRNA vaccines are effective in the treatment of malignant tumors; however, their efficacy in head and neck squamous cell carcinoma (HNSCC) has not been determined. This study aimed to identify the potential antigens of HNSCC for mRNA vaccine development and further distinguish the immune subtypes of HNSCC to select suitable patients for vaccination. Methods: We obtained gene expression profiles and corresponding clinical information of HNSCC from Gene Expression Omnibus (GEO) and The Cancer Genome Atlas (TCGA). We visualized the genetic alterations of potential antitumor antigens using cBioPortal and obtained the immune gene set from Immport. The correlation between the expression of the identified antigens and the infiltration of antigen-presenting cells was visualized by Tumor Immune Estimation Resource (TIMER). We evaluated the potential biological functions of different samples and described the immune landscape. Results: Increased expression of three potential tumor antigens, CCR4, TMCO1, and SPACA4, associated with superior prognoses and infiltration of antigen-presenting cells, was identified in HNSCC. Three immune subtypes (C1–C3) with different molecular, cellular, and clinical characteristics were defined. Patients with C3 tumor had a better prognosis, representing an immune “cold” phenotype, which may be more suitable for mRNA vaccination. In addition, different immune characteristics were observed among the three immune subtypes, including markers of immune cells, mutation burden, expression of immune checkpoints, and immune modulators. Finally, the immune landscape of HNSCC showed a high degree of heterogeneity between individual patients. Conclusion: CCR4, TMCO1, and SPACA4 may be potential antigens for developing mRNA vaccines against HNSCC, especially for patients with C3 tumor. This study could provide a theoretical basis for the development of an mRNA vaccine against HNSCC.

## 1. Introduction

Head and neck squamous cell carcinoma (HNSCC) is a lethal malignancy with biodiversity and genomic heterogeneity. HNSCC is the sixth most common malignant tumor in the world, with 600,000 newly diagnosed cases and 350,000 deaths every year [1]. In the past three decades, the survival rate of HNSCC has improved slightly with the advancement of medical technology and treatment; however, the five-year survival rate is still only about 66% [2]. Despite the approval of immune checkpoint blockade therapy for HNSCC, the overall response rate is less than 20% [3,4]. Therefore, novel approaches are needed to improve the current treatment strategies for HNSCC.

mRNA vaccines are a new technology combining molecular biology and immunology, which is closely related to gene therapy. In 1990, Wolff et al. reported, for the first time, that the corresponding protein products were detected after the direct injection of in vitro transcribed mRNA into mice [5]. This finding attracted attention to mRNA therapy. mRNA approaches have many advantages. First, the open reading frame (ORF) of the mRNA contains only the gene encoding the antigen. In addition, proper modification of mRNA components and 3ʹ-terminal untranslated region can significantly improve mRNA stability and translation efficiency [6,7,8]. With the development of delivery vehicles, the stability and translation efficiency of mRNA vaccines have also been significantly improved. Furthermore, the introduction of mRNA encoding MHC-I and tumor-associated antigen (TAA) into tumor cells can improve the immunogenicity of tumors and improve the therapeutic efficacy of mRNA vaccines [9,10]. Therefore, mRNA vaccines have tremendous potential in the fight against cancer and viral diseases due to their favorable safety profiles, efficacy, and ease of industrial production.

mRNA vaccines can induce innate and adaptive immunity [11]. They induce an increase of tumor infiltrating lymphocytes in the tumor microenvironment to achieve an anti-tumor effect. In recent years, an increasing number of studies have confirmed the safety and therapeutic effect of mRNA vaccines on different malignancies. For example, a nanoparticle-loaded MUC1 mRNA vaccine could induce potent cytotoxic T lymphocyte responses in triple-negative breast cancer [12]. Meanwhile, liposome-packaged mRNA encoding tumor-associated antigens gp100 and TRP2 could significantly induce toxic CD8^+^ T cell response and prolong the overall survival rate of melanoma mice [13]. Currently, multiple clinical trials of mRNA vaccines are being carried out for different malignancies and infectious diseases under different delivery conditions [11,14,15,16,17]. Considering the remarkable efficacy of mRNA vaccines, they may be promising candidates for tumor therapy. However, there are few studies on mRNA vaccines against HNSCC.

The purpose of this study was to identify potential tumor antigens from publicly available databases for the development of an anti-HNSCC mRNA vaccine and to determine the immune profile of HNSCC patients which would be suitable for vaccination via bioinformatics methods. In this study, we identified three tumor antigens, CCR4, TMCO1, and SPACA4, related to the prognosis and antigen-presenting cell infiltration of HNSCC. Based on the clustering of immune-related genes, HNSCC patients were divided into three immune subtypes and six functional modules with different cellular, molecular, and clinical characteristics. The C3 subtype, i.e., the immune cold subtype, had lower IFN-γ response, TGF-β response, macrophages, and T cells CD4 memory response relative to the C1 and C2 subtypes. This indicated that mRNA vaccination may be most effective with the C3 subtype. In addition, by defining the immune landscape of HNSCC, we illustrate the complexity of the immune microenvironment of HNSCC and provide a reliable reference for the further development and management of mRNA vaccines.

## 2. Materials and Methods

### 2.1. Data Acquisition

We obtained the RNA-seq data of HNSCC samples (including 502 HNSCC samples and 44 normal samples) from The Cancer Genome Atlas (TCGA, https://portal.gdc.cancer.gov/, accessed on 7 July 2021). The cBioPortal (http://www.cbioportal.org, version 3.2.11, accessed on 7 July 2021) was used to show genetic alterations in the underlying anti-tumor antigen in TCGA. The Series Matrix File of GSE39366 was downloaded from the Gene Expression Omnibus (GEO, http://www.ncbi.nlm.nih.gov/geo/, accessed on 7 July 2021) public database. The annotation platform was GPL9053. The immune gene set (including 1811 immune-related genes) was obtained from the ImmPort database. Our workflow is illustrated in Appendix A.

### 2.2. Tumor Immune Evaluation

The Tumor Immune Estimation Resource (TIMER, https://cistrome.shinyapps.io/timer/, accessed on 14 July 2021) is a comprehensive resource for the systematic analysis of the immune infiltrates of diverse cancer types. In this study, TIMER was used to visualize the relationship between the infiltration of antigen-presenting cells (APCs) and the expression of the identified potential antigens.

### 2.3. Classification of Immune Subtypes

The expression profiles of immune-related genes were clustered unsupervised using the “Nonnegative Matrix Factorization (NMF)” package [18]. Then, the expression levels of genes were normalized, and Cox regression analysis was performed to evaluate the correlation between candidate genes and overall survival (OS) using the “Survival” package [19]. Then, the unsupervised NMF clustering method was applied, using the “NMF” package, and the same candidate genes were used to apply the method to the external validation set of GEO (to make the expression profile of the validation set a non-negative matrix, 8 was added to the expression values). The k value, whose correlation coefficient began to decline, was selected as the best cluster number. Then, based on the T-SNE approach, the mRNA expression data of the above immune genes were used to verify the subtype distribution.

### 2.4. Drug Sensitivity Analysis

Drawing on the largest existing drug database (GDSC cancer drug sensitivity genomics database, https://www.cancerrxgene.org/, accessed on 14 July 2021), we used the “pRRophetic” package [20] to predict the chemotherapy sensitivity of each tumor sample. Estimates of the IC50 of each chemotherapeutic agent were obtained by regression, and the regression and prediction accuracy was tested by 10 cross-validation tests using the GDSC training set. Default values were selected for all parameters, including “combat” to remove batch effects and the average of repeated gene expression.

### 2.5. Gene Set Variation Analysis

Gene set variation analysis (GSVA) [21] is a nonparametric and unsupervised method to evaluate the enrichment of transcriptome gene sets. In this study, gene sets were downloaded from the Molecular Signatures Database (Version 7.0) (https://www.gsea-msigdb.org/gsea/msigdb/index.jsp, accessed on 14 July 2021), and aggregated scores were given for each gene set using the GSVA algorithm to assess potential biological functional changes in different samples.

### 2.6. Immune Landscape Analysis

The Reduce Dimension function of the Monocle package [22] with normal distribution was adopted in our dimension reduction analysis. The maximum number of components was set to 4, and the discriminant tree dimension reduction algorithm (DDRTree) [23] was used to reduce the dimension. Finally, the plot_cell_trajectory function (package Monocle) was used to visualize the immune landscape.

### 2.7. Weighted Gene Co-Expression Network Analysis (WGCNA)

By constructing a weighted gene co-expression network, the co-expression gene modules were found, and the associations between the gene network and phenotype were explored, as well as the core genes. The “WGCNA” R software package [24] was used to construct a co-expression network of all the genes in the data set, and the top 5000 genes with variance were screened by this algorithm for further analysis, in which the soft threshold was set to 3. The weighted adjacency matrix was transformed into a topological overlap matrix (TOM) to estimate the network connectivity, and the hierarchical clustering method was used to construct the cluster tree structure of the TOM matrix. Based on their weighted correlation coefficients, genes were classified according to their expression patterns. Those with similar patterns were grouped into a module, and tens of thousands of genes were divided into multiple modules through gene expression patterns.

### 2.8. Analysis of GO and KEGG Functions

Functional annotation of key genes was performed using the “clusterProfiler” package (R3.6) [25] to fully explore the functional correlations of the candidate genes. The Gene Ontology (GO) and Kyoto Encyclopedia of Genes and Genomes (KEGG) were used to assess related functional categories. GO and KEGG enrichment pathways with both *p* and *Q* values < 0.05 were considered significant pathways.

### 2.9. Subtype GSEA Analysis

The log2FC (Fold Change) value for each gene between the subtypes was obtained using the “limma” [26] and “clusterProfiler” [25] packages, while the GO and KEGG pathways were analyzed using Gene Set Enrichment Analysis (GSEA) [27]. Finally, specific up-regulated pathways (10 pathways with the highest NES value) were selected from the results of each subtype, based on the MSigDB database (http://www.gsea-msigdb.org/gsea/downloads.jsp, accessed on 14 July 2021).

### 2.10. Statistical Analysis

A survival analysis was carried out using Kaplan Meier methods and the results were compared using the log-rank test. The Cox proportional risk model was used for multivariate analysis. All statistical analyses were performed using the R language (Version 3.6, The University of Auckland, Auckland, New Zealand). All statistical tests were two-sided, with *p <* 0.05 indicating statistical significance.

## 3. Results

### 3.1. Identification of Immune-Critical Genes in HNSCC

Through the cBioportal online tool, we obtained 21,999 genes that had copy number variations and 13,304 genes that had mutations in TCGA-HNSCC (Figure 1A,B). Among them, the overall survival (OS) analysis of 4136 genes was significant (*p* < 0.05), and the disease-free survival (RFS) analysis of 2844 genes was significant (*p* < 0.05). On this basis, three genes, i.e., CCR4, TMCO1, and SPACA4, were retained (Figure 1C); their expressions were significantly correlated with the OS and RFS of HNSCC (Figure 1D–I). B cells, macrophages, and dendritic cells (DCs) were the major antigen-presenting cells (APCs). Furthermore, CCR4 was significantly correlated with tumor infiltration of B cells, CD8^+^ T cells, CD4^+^ T cells, and macrophages, while TMCO1 was significantly correlated with CD8^+^ T cells, and SPACA4 was significantly correlated with B cells, CD8^+^ T cells and Neutrophils (Figure 2A–C). The above results indicate that CCR4, TMCO1, and SPACA4 have potential immune stimulating properties and could be processed by APCs to induce an anti-tumor immune response.

### 3.2. Identification of Potential Immune Subtypes of HNSC

To further identify the key genes in the candidate gene set, we collected the clinical information of HNSCC patients, and screened the characteristic genes in HNSCC by Cox univariate regression analysis, allowing us to filter out 111 prognostic genes (*p* < 0.01). We clustered the TCGA data set containing HNSCC samples according to the expression profiles of these 111 candidate genes using the NMF consensus clustering method, and selected k = 3 as the best cluster number (Figure 3A). Subsequently, the data set of the GSE39366 cohort was independently verified. Using the previously selected k = 3 classification, three different molecular subtypes were revealed, showing significant prognostic differences in the TCGA dataset. Compared with the C1 and C2 subtypes, the overall prognosis of the C3 subtype was better (Figure 3B). Similar differences were observed in the GSE39366 subtype (Figure 3C), indicating the stability and reproducibility of the results.

We then analyzed the immune status of three subtypes. It was found that the C3 subtype had lower IFN-γ response, TGF-β response, macrophages, and T cells CD4 memory response relative to the C1 and C2 subtypes (Figure 4A), belonging to the immune cold subtype. This indicated that mRNA vaccination might be most effective with the C3 subtype. Considering the therapeutic effect of HNSCC by chemotherapy, we predicted the chemosensitivity of each tumor sample to further explore the sensitivity of common chemotherapy drugs based on the immune subtypes. The results showed that the immune subtypes were significantly correlated with the sensitivity of patients to bexarotene, CMK, dasatinib, docetaxel, metformin, and mitomycin. C (Appendix A). We further explored the mutation map of patients according to immune subtypes (Figure 4B); this showed that the mutation proportion of multiple genes, such as TP53, TTN, and FAT1, in the high-risk group was significantly different among subtypes. Unfortunately, differences in the tumor mutational burden (TMB) among the expression groups of the various subtypes did not achieve statistical significance (Appendix A).

### 3.3. Molecular Characteristics of Immune Subtypes

We analyzed the expression of immune checkpoints, immunomodulators, and cell report gene among subtypes. It was found that the expression of a large number of immune checkpoints and immunomodulatory genes was significant among subtypes (Figure 5A–E), suggesting that the abnormal disturbance of immunomodulatory pathways among the three subtypes is the potential mechanism explaining the difference in prognosis among the three groups of patients. In a previous study [28] based on the immunogenomic analysis of more than 1000 tumor samples from 33 cancers, Thomson et al. identified six immune categories (*c_1_*–*c_6_*) which were significantly correlated with tumor prognosis and genetic and immunomodulatory changes. To prove the reliability of our immunotyping, the correlation between three immune subtypes of HNSCC and the six immune subtypes (*c_1_*–*c_6_*) was explored (Figure 6A). It was found that HNSCC was mainly concentrated in *c_1_*, *c_2_*, *c_3_*, *c_4_*, and *c_6_*, of which *c_1_* (wound healing) was distributed in C3, while *c_3_* (infrared) and *c_4_* (lymphocytic completion) were mainly distributed in C2. This shows that the immune microenvironment of HNSCC is significantly different from those of other tumor types.

In addition, the expression differences of immune cell markers between the three subgroups were analyzed (Figure 6B–E). There were significant differences in the expression of cell markers from activated CD8^+^ T cells, activated dendritic cells, macrophages, and NK cells among the three subgroups. Next, through a quantitative analysis using the single sample GSEA (ssGSEA) algorithm, it was found that there were many related pathways with obvious differences among the three subgroups (Appendix A), among which the pathways with higher scores in subtype C3 in GO analysis were URETE_METABOLIC_PROCESS, 3_UTR_MEDIATED_mRNA_STABILIZATION, and TRANSFORMATION_ACTIVATOR_ACTIVITY (Appendix A). In our KEGG analysis, the pathways with highest scores in subtype C3 were METABOLISM_OF_XENOBIOTICS_BY_CYTOCHROME_P450, ASCORBATE_AND_ALDARATE_METABOLISM, and drag_METABOLISM_CYTOCHROME_P450 (Appendix A).

### 3.4. Immune Landscape of HNSCC

Then, immune-related gene expression profiles were integrated to construct the immune landscape of HNSCC in order to visualize the immune distribution of each patient (Figure 7A). The distribution of our three immune subtypes in the immune pattern was not consistent (Figure 7B,C). PCA1 was found to be related to a variety of immune cells, including the activation of macrophage M0 and DCs, while PCA2 was most negatively related to the activation of NK cells. We further analyzed the prognoses of widely distributed patients and found that the those of the S1 subtype were better than those of the S6 and S7 subtypes (Figure 7D,E). According to the distribution position of the three immune subtypes in the immune pattern, C1, C2, and C3 were further divided into different subgroups (Figure 7F). We further analyzed the survival of the subgroups assigned to each subtype (Figure 7G–I) and found that the prognosis of the C1-1 subtype was better than that of C1-7, while the prognosis of C3-1 was significantly better than that of C3-6; however, there was no significant difference in prognoses in the C2 subtypes.

### 3.5. HNSCC Immune Gene Co-Expression Module

To determine the co-expression network of immune-related genes in the HNSCC cohort, we performed WGCNA analysis. We took the immune subtypes C1, C2, and C3 as the clinical traits of the samples and further used them to construct a WGCNA network to explore the biomarkers in HNSCC. A sample clustering diagram is shown in Appendix A. Soft threshold β was set to 3, as determined by the sft$powerEstimate function (Figure 8A). Then, six gene modules were detected based on the Tom matrix, i.e., blue (n = 225), brown (n = 98), green (n = 71), turquoise (n = 652), yellow (n = 94), and green (n = 60) (Figure 8B–D). Through further analysis of modules and traits, we found that the correlation between the MEblue module and sample category (immune subtype) was the highest and was negatively correlated (Cor = −0.24, *p* < 0.01) (Figure 8E). Therefore, the MEBlue module was selected for subsequent correlation verification analysis. The modular characteristic genes of the three immune subtypes were analyzed, and the expression of the characteristic genes in blue, brown, grey, and turquoise modules was significant among subtypes (Figure 8F,G).

It was found that the genes of the MEblue module were significantly enriched in ameboidal-type cell migration, focal adhesion, and receiver ligand activity by GO enrichment analysis (Figure 9A), and the TGF-β and HIF-1 signaling pathways by KEGG enrichment analysis (Figure 9B). At the same time, we analyzed the protein–protein interaction (PPI) network of the genes in the modeling candidate gene set using the Cytoscape software (Cytoscape, https://cytoscape.org/, accessed on 15 November 2022, Cytoscape Consortium, San Francisco, CA, USA) (Figure 9C). Next, we studied the specific signal pathways that may be related to the genes of the MEblue module and explored the potential molecular mechanisms that may affect the lesion and progression of head and neck cancer by Gene Set Variation Analysis (GSVA) (Figure 9D). This showed that different immune subtype groups were mainly enriched in the differential pathways.

In addition, a prognostic correlation analysis showed that only gene expression in the meturquoise module was significantly correlated with the prognosis of HNSCC patients (Figure 10A). Among them, the MEbrown, MEgreen, MEturquoise, and MEyellow modules were significantly negatively correlated with PCA1, while the MEBlue, MEbrown, MEgreen, MEyellow, and MEturquoise modules were significantly positively correlated with PCA2 (Figure 10B–M).

## 4. Discussion

HNSCC is one of the most common malignant tumors in the world. At present, surgical treatment, chemoradiotherapy, or combined therapy are still common treatment methods. Clinical trials of immune checkpoint inhibitors have proved the clinical efficacy of pembrolizumab and nivolumab [29,30], but the overall response rate of HNSCC is less than 20% [3,4]. At present, many clinical trials are exploring new immunotherapies. As a new type of immunotherapy, mRNA tumor vaccines have achieved significant curative effects in preclinical models and clinical studies of different malignancies including melanoma [13,31], breast cancer [12], colorectal cancer [32], and gastrointestinal cancer [33]. However, there is little research on the use of mRNA tumor vaccines with HNSCC.

In this study, we constructed the copy number alterations and mutation landscape of HNSCC and identified a series of targetable antigens. Three tumor antigens (CCR4, TMCO1, and SPACA4) related to APCs infiltration and prognosis in HNSCC were identified, and it is expected that these genes might become candidate antigens for an mRNA vaccine. However, the specific effects in HNSCC need to be further verified by functional experiments. Reports in the literature pave the way for these three tumor antigens to become mRNA vaccines against HNSCC. For example, studies have suggested that CCR4 is significantly associated with the tumor microenvironment (TME) and prognosis in HNSCC [34,35], and that CCR4 may serve as a new potential molecular target for HNSCC therapy [36]. It should be noted that further research is essential, and there is an urgent need to validate the effects of these identified vaccine antigens in HNSCC.

Considering that the mRNA vaccine is only effective for some patients, we clustered HNSCC patients into three immune subtypes to identify those that may be suitable for vaccination. These three subtypes have different clinical characteristics. Compared with C1 and C2 subtypes, the C3 subtype has a better survival prognosis, which indicates that immune typing can be used to predict the prognosis of HNSCC patients. The C3 subtype belongs to the immune cold subtype. Cancer vaccines as adjuvants directly activate APC and expand the tumor-specific T-cell repertoire, expand the tumor-specific T-cell repertoire, and enhance the innate anti-tumor immunity of the host [37]. mRNA vaccines can change the tumor microenvironment from “cold” tumors to “hot” tumors, thus killing the tumors [38,39]. Therefore, we speculate that mRNA vaccines may be most effective with the C3 subtype. However, one study of 289 H&E whole slides of HNSCC found that immunologic “hot” and “excluded” HNSCCs were associated with better overall survival than “cold” HNSCC patients [40]. This result is in contrast to the findings in our study, perhaps due to differences in the assessment of immune status. The authors of [40] distinguished three categories by reading H&E slides, while in our study, the evaluation of “cold” and “hot” tumors was based on a comprehensive evaluation of the tumor microenvironment by GSVA. The C3 subtype has a lower TP53 mutation load and better survival prognosis, which is consistent with the previous literature results [41]. Similarly, a study assessing the risk of the immune microenvironment in HNSCC demonstrated that the lower the myeloid gene score, the better the response to immunotherapy and the better the clinical prognosis [42]. Some studies using systems of unsupervised classification based on tumor-related variable splicing genes have found that four categories exist, with different molecular and clinical characteristics [43]. Combined with this study, this further explains the complexity of the internal environment of the tumor, which is one of the challenges associated with designing personalized mRNA vaccines.

In addition, immune subtypes also reflect the expression levels of immune checkpoints and immune modulators. The up-regulated expression of immune modulators is more suitable for a vaccination with an mRNA vaccine, while the high expression of immune checkpoints is not suitable. Our analysis results show that the expression levels of immunosuppressants *TGFB1, TGFBR1*, *IL10*, and *PDCD1LG2* are relatively high among the C1 types, which may be one of the reasons for the poor prognoses associated with this subtype. This is consistent with the results reported previously; for example, *TGFB* family genes have been significantly associated with decreased survival of HNSCC patients [44]. Additionally, according to an analysis of the molecular profiling of circulating tumor cells (CTCs) and clinical factors in HNSCC, it was found that the expression of *PDCD1LG2* as PD-1 ligand genes was related to poor prognosis of HNSCC [45]. However, the expression levels of cytokines CXCL8, CXCL1, CXCL6, and CXCL17 in the C2 subtype are higher, while the prognosis of the C2 subtype is relatively poor compared with those of the C1 and C3 subgroups. This is consistent with the results reported previously [46] and may be related to tumor heterogeneity, suggesting that the abnormal disturbance of immune regulatory pathways among the three subtypes of HNSCC may be the potential mechanism of the difference in prognosis among the three groups.

We studied the distribution of six categories (*c_1_*–*c_6_*) reported previously [28] and observed that our three subtypes of HNSCC have different distribution rates, indicating that the immune subtype of HNSCC reported by us is different from the previously determined immune subtype and other tumor subtypes, such as cholangiocarcinoma [47] and pancreatic adenocarcinoma [48]. Therefore, our results could supplement the classification of the immune microenvironment of HNSCC. Meanwhile, further dimensionality reduction revealed that the heterogeneity in the C2 subgroup is not significant. This may be related to different dominant immune factors that significantly affect the prognosis of patients. More research is needed to better understand the tumor microenvironment. The complex immune landscape of HNSCC also showed that there was considerable heterogeneity between individual patients and within the same immune subtype, which reduces the immune components which could be used in the development of personalized gene vaccine-based therapies. In these patients, a combination of an mRNA-based cancer vaccine with another immunotherapy or chemotherapy may prove to more effectively regulate the host immune response and the tumor microenvironment. Various reviews have summarized the importance of cancer vaccines and emphasized that vaccine-induced immune initiation produces a set of effectors whose function can be enhanced by immune checkpoint receptor (ICR) inhibitors [37,49].

However, there are some limitations in this study. First, we lack multi-center clinical data to validate our conclusions. In addition, the specific effects of the three tumor vaccines in HNSCC need to be further verified by functional experiments, and the application of mRNA vaccine in HNSCC needs further research. Additionally, a vaccine alone is less likely to achieve long-term tumor regression, and therefore, cancer vaccine design, antigen, adjuvant, and delivery methods need to be further explored to improve cancer vaccines. Given the effect of the mRNA vaccine in coronavirus disease-2019 (COVID-19), this study provides important information for mRNA vaccine development for other diseases.

## 5. Conclusions

mRNA vaccines represent a promising tumor immunotherapy method. This study determined that CCR4, TMCO1, and SPACA4 may be potential antigens for HNSCC mRNA tumor vaccine development, and patients with immune subtype C3 might benefit most from mRNA vaccination. Therefore, this study provides a theoretical basis for the mRNA vaccine against HNSCC.

## Figures and Tables

**Figure 1 biomolecules-13-00090-f001:**
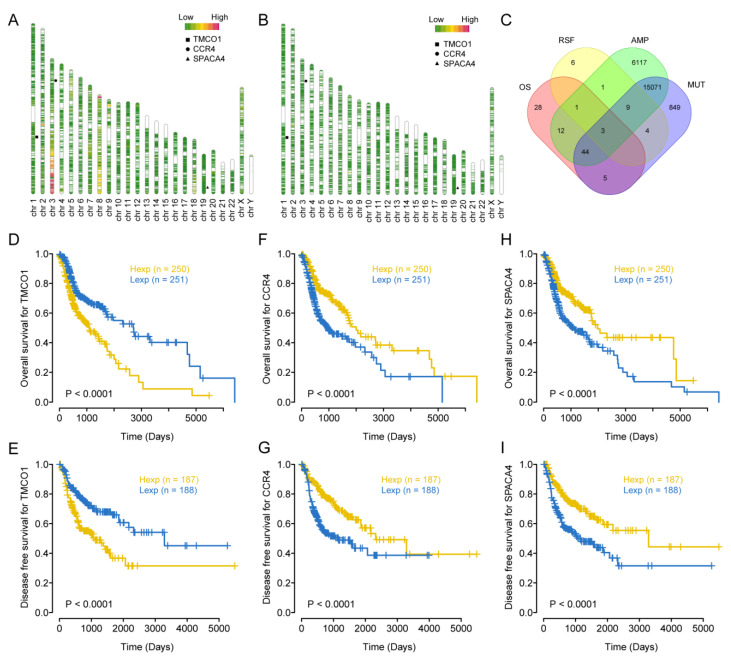
Identification of potential tumor antigens in HNSCC. (**A**) The map of amplification in HNSCC. (**B**) Map of mutation (MUT) in HNSCC. (**C**) Narrow-down analysis of potential tumor antigens with both amplified and mutated features, and significant OS and RFS prognosis (in a total of three candidates) in HNSCC. AMP represented amplification. Kaplan-Meier OS (**D**) and RFS (**E**) curves comparing the groups with a different TMCO1 expression in HNSCC. Kaplan-Meier OS (**F**) and RFS (**G**) curves comparing the groups with a different CCR4 expression in HNSCC. Kaplan-Meier OS (**H**) and RFS (**I**) curves comparing the groups with a different SPACA4 expression in HNSCC. Hexp and Lexp represented high expression and low expression, respectively.

**Figure 2 biomolecules-13-00090-f002:**
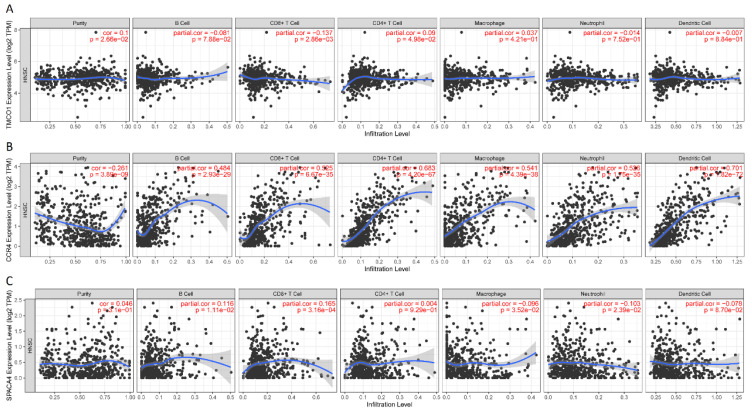
Identification of tumor antigens associated with antigen-presenting cells. Association of TMCO1 (**A**), CCR4 (**B**), SPACA4 (**C**) expression with the purity of infiltrating cells and numbers of B cells, CD8^+^ T cells, CD4^+^ T cells, macrophages, neutrophils, and dendritic cells in HNSCC.

**Figure 3 biomolecules-13-00090-f003:**
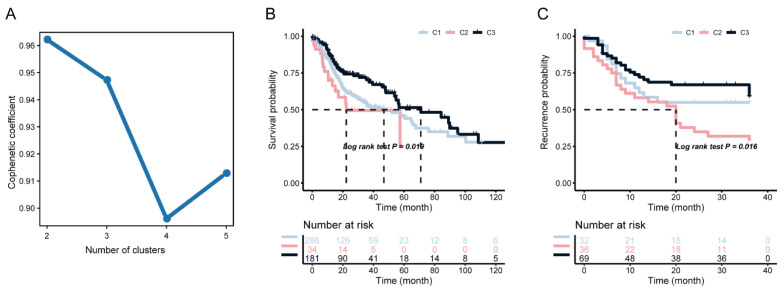
NMF consensus clustering identifies potential immune subtypes in HNSC. (**A**) The cophenetic coefficient for NMF in TCGA, K = 2–5. Survival analysis of the immune subtypes in the TCGA cohort (**B**) and GEO cohort (**C**), respectively.

**Figure 4 biomolecules-13-00090-f004:**
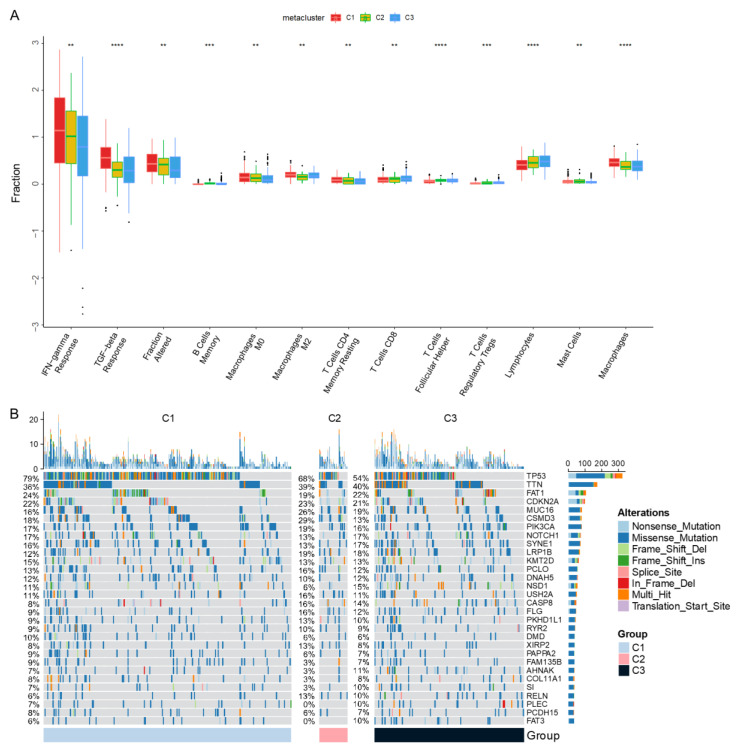
Multiomics analysis of immune subtypes. (**A**) Analysis of immune-related molecular characteristics among immune subtypes. (**B**) The landscape of the genomic alteration of immune subtypes. ** *p* < 0.01, *** *p* < 0.001, and **** *p* < 0.0001.

**Figure 5 biomolecules-13-00090-f005:**
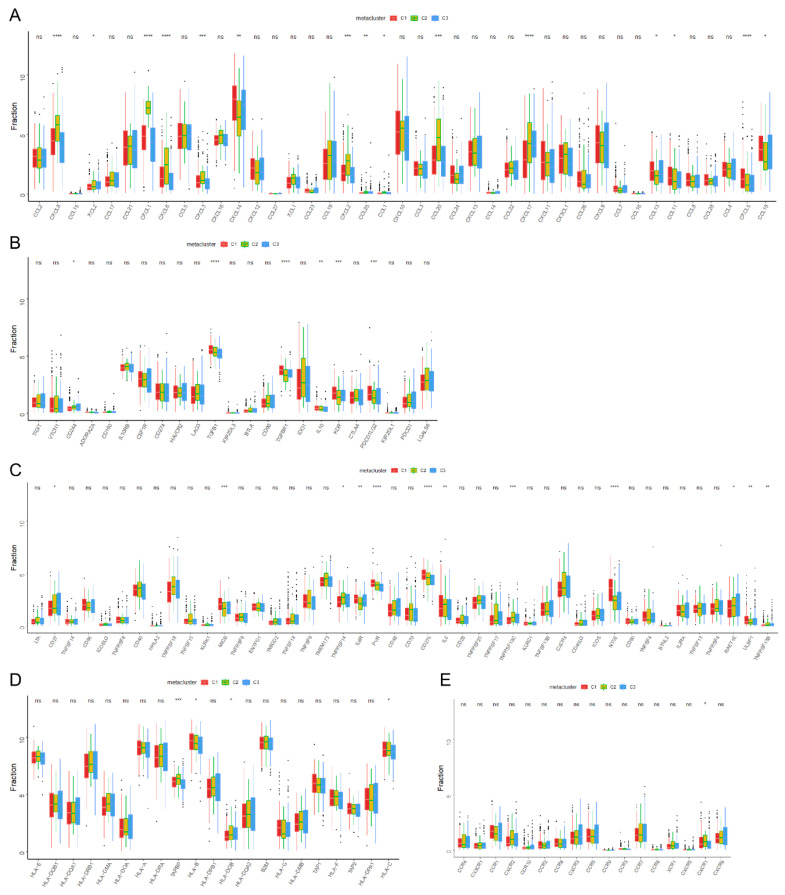
Analysis of immunomodulator and chemokine-related genes among immune subgroups. Differential expression analysis of chemokine-related genes (**A**), immunosuppressant-related genes (**B**), immunostimulator-related genes (**C**), MHC-related genes (**D**), and receptor-related genes (**E**) among immune subtypes. * *p* < 0.05, ** *p* < 0.01, *** *p* < 0.001, and **** *p* < 0.0001.

**Figure 6 biomolecules-13-00090-f006:**
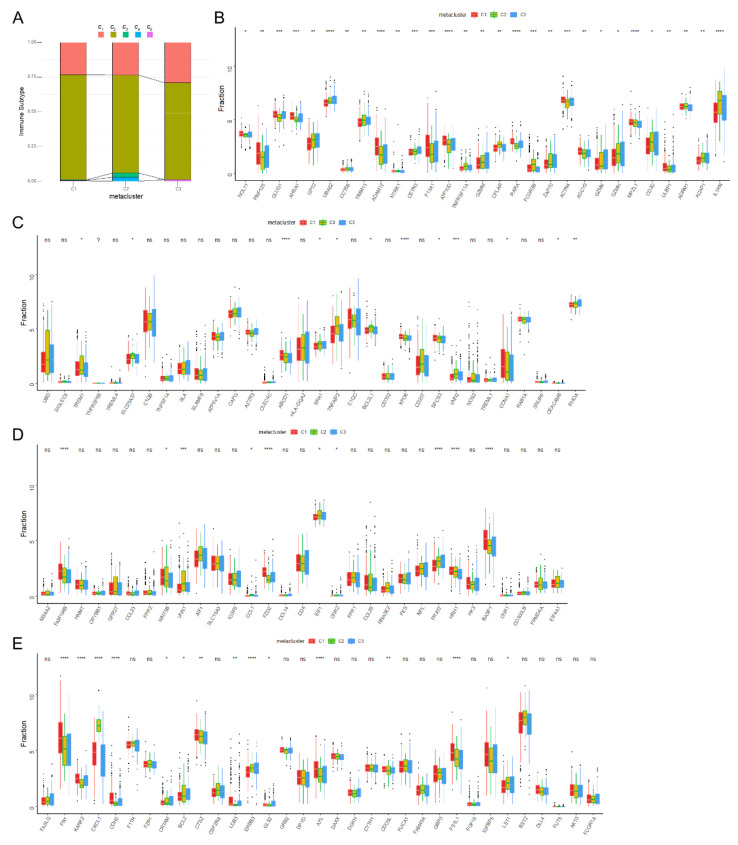
Cellular and molecular characteristics of immune subtypes. (**A**) The correlation between three immune subtypes (C1–C3) from our results and six immune subtypes (*c_1_*–*c_6_*) from Thomson. Differential expression analysis of activated CD8 T cell-related genes (**B**) and activated dendritic cell-related genes (**C**). Macrophage-related genes (**D**) and NK cell-related genes (**E**) among subtypes. * *p* < 0.05, ** *p* < 0.01, *** *p* < 0.001, and **** *p* < 0.0001.

**Figure 7 biomolecules-13-00090-f007:**
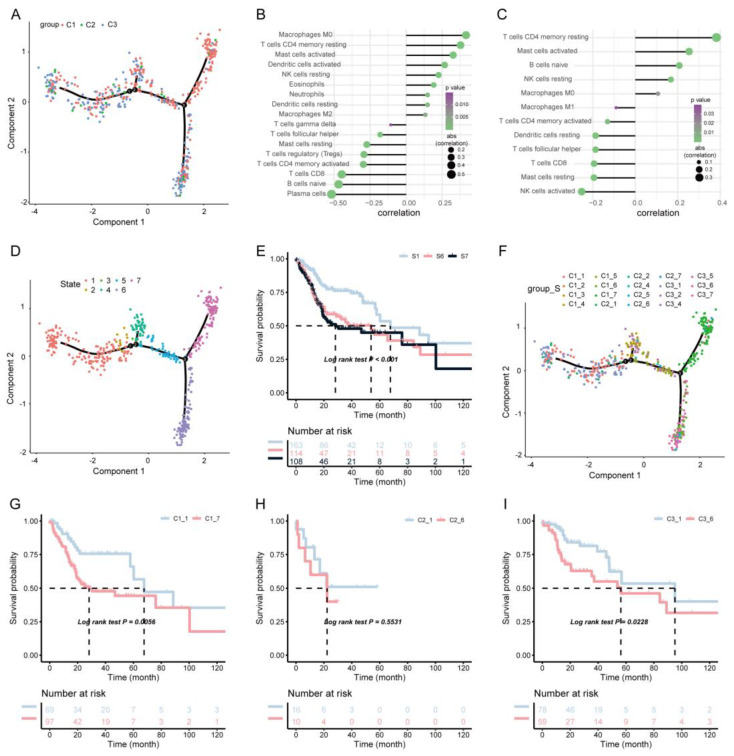
The immune landscape of HNSC. (**A**) The immune landscape of HNSC. Each point represents a patient, and the immune subtypes are color-coded. The horizontal axis represents the first principal component, and the vertical axis represents the second principal component. (**B**) Correlation between PCA1 and immune infiltration. (**C**) Correlation between PCA2 and immune infiltration. (**D**) Patients were separated by the immune landscape based on their location; (**E**) separated patients were associated with different prognoses. (**F**) C1, C2, and C3 are further stratified according to their positions in the immune environment. Different groups in C1 (**G**), C2 (**H**), and C3 (**I**) were associated with different prognoses.

**Figure 8 biomolecules-13-00090-f008:**
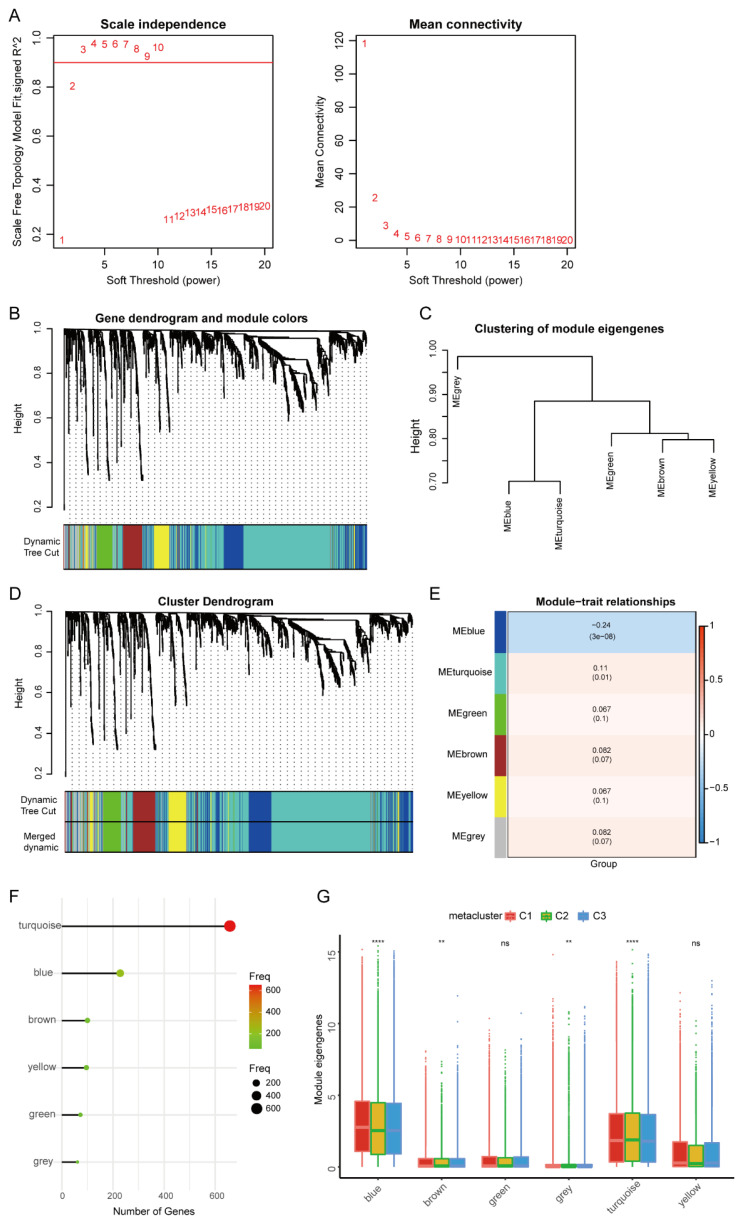
Identification of immune gene co-expression modules of HNSC. (**A**) Scale-free fit index for various soft-thresholding powers (β), and mean connectivity for various soft-thresholding powers. (**B**) Dynamic tree of feature modules. (**C**) Clustering tree of feature modules. (**D**) Dendrogram of all differentially expressed genes clustered based on a dissimilarity measure (1-TOM). (**E**) Relationship between modules and immune subtypes. (**F**) Gene numbers in each module. (**G**) Differential distribution of feature vectors of each module in HNSC subtypes. Expression of the identified gene modules in the immune subtypes. ** *p* < 0.01 and **** *p* < 0.0001.

**Figure 9 biomolecules-13-00090-f009:**
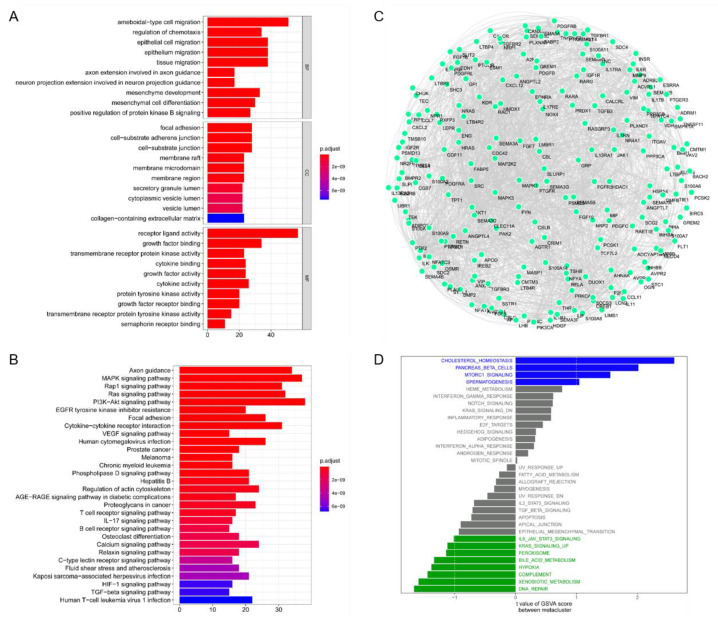
Functional analysis of key module genes. (**A**) Go enrichment analysis. (**B**) KEGG enrichment analysis. (**C**) PPI network analysis. (**D**) GSVA analysis.

**Figure 10 biomolecules-13-00090-f010:**
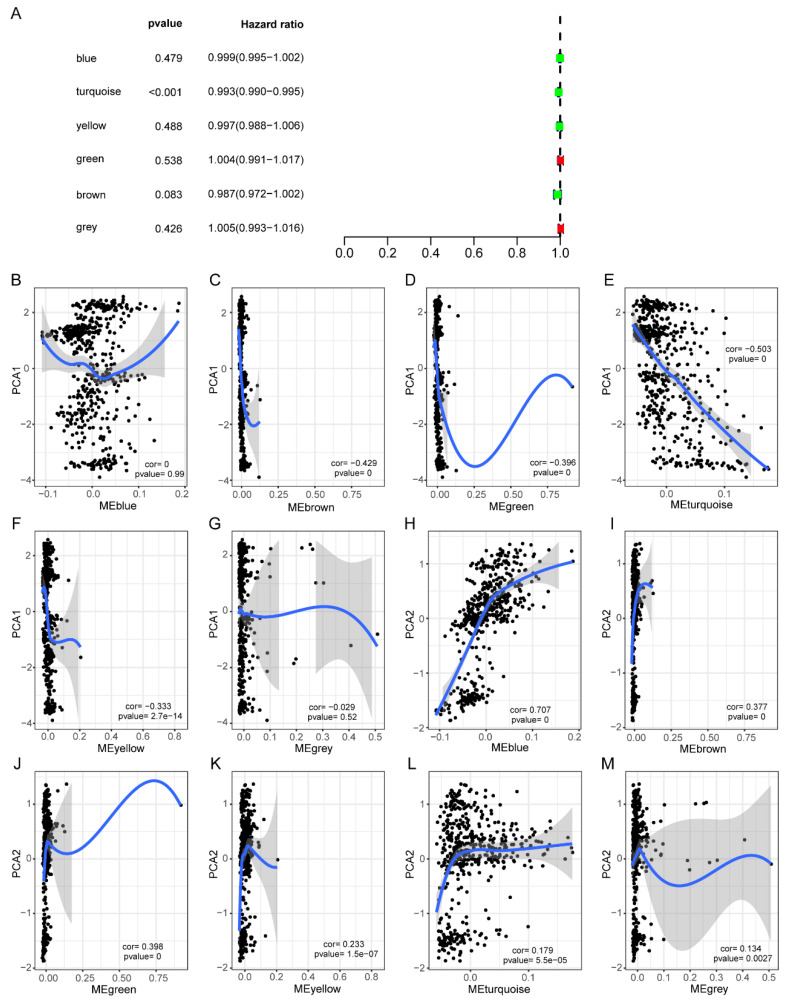
Identification of key immune genes of HNSC. (**A**) Forest maps of single-factor survival analysis of six modules of HNSC. (**B**–**G**) Correlation between six module feature vector and PCA1 in immune landscape. (**H**–**M**) Correlation between six module feature vector and PCA2 in immune landscape.

## Data Availability

All data generated and described in this article are available from the corresponding web servers. The datasets analysed during the current study are available in the TCGA repository (https://portal.gdc.cancer.gov/, accessed on 7 July 2021), GEO (http://www.ncbi.nlm.nih.gov/geo, Accession No. GSE39366, accessed on 7 July 2021), cBioPortal (http://www.cbioportal.org, version 3.2.11, accessed on 7 July 2021).

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
