# Peer review of "Screening of Tumor Antigens and Construction of Immune Subtypes for mRNA Vaccine Development in Head and Neck Squamous Cell Carcinoma"

_biomolecules, 2022, doi:10.3390/biom13010090_

Round 1
Reviewer 1 Report
The manuscript by Li et al. attempted to identify tumor antigens for mRNA vaccine development for HNSCC. The authors identified 3 potential tumor antigens, CCR4, TMCO1, and SPACA4, for mRNA vaccine development, and 3 immune subtypes (C1- C3), C3 of which may be more suitable for mRNA vaccination. The authors conducted thorough analyses and presented some interesting results. In the backdrop of the successful development of COVID mRNA vaccines, this paper promises to further expand the current capability of mRNA-based therapeutics. Therefore, we recommend the acceptance of this paper, but the following concerns as listed below for the authors to improve their paper.
1) In the introduction (line 50), the authors suggested “new treatment methods are needed to improve”. However, mRNA vaccine appears to prevent the occurrence of the diseases, but not to treat the diseases. Could the authors further clarify this point?
2) The data presented in Results (session 3) should be presented in a way so that the relevance to mRNA vaccine development can be better manifested. Currently, the data appear mostly irrelevant to mRNA vaccine. For example, Fig. 4B shows the drug sensitivity analysis among immune subtypes, but we could not get how this information can help mRNA vaccine development.
3) Some abbreviations and figure legends need explanation. For example, TAA in line 61, AMP in Fig. 1C, Hexp & Lexp in Fig. 1D-I, TMB in line 220, TME in line 346, etc.
4) Most figures need more explanations; perhaps more detailed captions would be helpful. For example, for Fig. 2, we could not discern how to determine whether or not a certain antigen is “correlated” with a certain type of APC.
5) Some figures are extremely difficult to interpret due to low-res or inappropriate presentation. For example, Fig. 9C is a big cloud with no discernible information. For Fig. 7A, D, F, why both the axes are “component 2”?
6) Grammar mistakes or typos: line 211, “which is indicated that” should be “which indicated that”; “Dendritic cell” should be “dendritic cell”, etc.
Author Response
Thank you very much for allowing us to revise our manuscript (biomolecules-2069744). We have read carefully the reviewers' comments and found that they are very constructive. According to their suggestions, we have substantially revised the corresponding texts, especially in the “Results” and “Introduction”.
As for your comments, in the “Results” section, we have made detailed corrections to the comments by the reviewers and checked the full text for similar issues. In addition, the section “drug sensitivity analysis” was not the focus of this study, our study showed statistical differences in drug sensitivity among different subtypes, so we put this section into the "supporting information".
All revisions are marked in red. In the following pages are our point-by-point replies to the comments of the reviewers.
Thank you for your time and consideration! With all the best,
Prof. Wei Sun

Reviewer 2 Report
Dear Authors
It was with great pleasure that I reviewed your manuscript.
I have a few comments to make:
The main objective of this study is to identify the potential HNSCC antigens for mRNA vaccine development, and further to distinguish the immune subtypes of HNSCC to select the patients for vaccination.
Do you consider the topic original or relevant in the field? Does it address a specific gap in the field?
The topic is relevant but not very original.
This study could provide a theoretical basis for the development of mRNA vaccine against HNSCC.
The methodology applied was the one indicated for this type of study.
The main problem of this study is related to the introduction which should be improved because this topic deserves to be treated in a more specific way.
My Best Regards
Author Response
Thank you very much for allowing us to revise our manuscript (biomolecules-2069744). We have read carefully the reviewers' comments and found that they are very constructive. According to their suggestions, we have substantially revised the corresponding texts, especially in the “Results” and “Introduction”.
As for the comments of Reviewer #2, firstly, we thank the reviewer for the affirmation of this study. In this study, we screened potential mRNA vaccines for HNSCC through public databases and analyzed the immune subtypes of HNSCC using bioinformatics methods to demonstrate the heterogeneity of HNSCC. We speculated that mRNA vaccines could only be used in some of HNSCC patients, for example, mRNA vaccines may be more suitable for C3 subtype. Therefore, the study is original. Secondly, we have made appropriate supplements in the “Introduction” section to make the study more substantial.
All revisions are marked in red. In the following pages are our point-by-point replies to the comments of the reviewers.
Thank you for your time and consideration! With all the best,
Prof. Wei Sun
